# Quality of Life Indicators in Patients Operated on for Breast Cancer in Relation to the Type of Surgery—A Retrospective Cohort Study of Women in Serbia

**DOI:** 10.3390/medicina56080402

**Published:** 2020-08-11

**Authors:** Predrag Kovačević, Snežana Miljković, Aleksandar Višnjić, Jefta Kozarski, Radmilo Janković

**Affiliations:** 1Department of Surgery, Faculty of Medicine, University of Niš, 18000 Niš, Serbia; predrag.kovacevic@medfak.ni.ac.rs (P.K.); drpredrag.kovacevic@gmail.com (R.J.); 2Clinic for Plastic and Reconstructive Surgery, Clinical Centre of Niš, 18000 Niš, Serbia; 3Department of Health Care, Faculty of Medicine, University of Niš, 18000 Niš, Serbia; snezana.miljkovic@medfak.ni.ac.rs; 4Center for Analysis, Planning and Organization, Clinical Centre of Niš, 18000 Niš, Serbia; 5Department of Social Medicine, Faculty of Medicine, University of Niš, 18000 Niš, Serbia; 6Institute of Public Health of Niš, Center of Analyzing, Planning and Organization of Health Care, 18000 Niš, Serbia; 7Department of Plastic Surgery, Faculty of Medicine, Medical Military Academy, 11000 Belgrade, Serbia; jeftakozarski@yahoo.com; 8Clinic for Anesthesia and Intensive Care, Clinical Centre of Niš, 18000 Niš, Serbia

**Keywords:** breast cancer, surgery, quality of life, WHOQOL-Bref, FACT-B

## Abstract

*Background and objectives:* Quality of life (QoL) after breast cancer surgery is an important public health issue. The aim of this study was to determine the relationship between the levels of perceived quality of life in patients operated on for breast cancer in relation to the type of surgery, using the standardized questionnaires. *Materials and Methods:* We assessed 425 women after surgery for breast cancer. The assessment included the application of the WHOQOL-bref (The World Health Organization Quality of Life-Bref), and FACT-B (Functional Assessment of Cancer Therapy-Breast) questionnaires. The statistical analysis of the data included multiple linear regression and correlation tests. *Results:* Multiple linear regression analysis found that education, existence of comorbidities, time elapsed since surgery, and type of surgery were significant predictors of overall quality of life. Women’s overall quality of life and general health has increased by 0.16 times for each subsequent year of surgery, and by 0.34 times for each subsequent higher education level. Breast-conserving surgery or mastectomy with breast reconstruction were statistically significant (*β* = 0.18) compared to total mastectomy. *Conclusions:* There is a significant difference in the quality of life perceived by patients in whom the breast has been preserved or reconstructed in relation to patients in whom total mastectomy has been performed.

## 1. Introduction

Breast cancer is a significant public health issue, and it is considered the second most common cause of death among women worldwide [1]. Worldwide, there were an estimated 2.1 million newly diagnosed female breast cancer cases in 2018, accounting for almost 1 in 4 cancer cases among women. The 5-year prevalence was nearly 7 million cases worldwide. It is also the leading cause of cancer death in over 100 countries [1].

Quality of life is an important parameter for assessing a patient’s health. It is well known that health is not only the absence of disease, but also mental well-being. A large number of chronic diseases and injuries leave consequences in terms of reduced overall life and work ability, as well as some other parameters of quality of life. This certainly includes the quality of life in patients operated on for breast cancer.

Nowadays, treatment protocols for these patients provide the possibility to perform surgery to remove only a part of the breast in strictly defined indications, as well as radiation therapy, which preserves an important organ of psychosocial identity for a woman. Unfortunately, in a larger number of patients, it is necessary to remove the entire breast tissue (mastectomy with axillary dissection). In this situation, a woman may feel stigmatized and withdrawn, resulting in impaired social functioning [2,3,4,5]. Surgical breast reconstruction can alleviate this problem and allow a woman suffering from cancer to suffer less emotional consequences of losing breast tissue.

In Serbia, health insurance is provided for all people, so the breast reconstruction is offered to patients every time, and it can be done immediately or secondarily.

After surgical treatment, some patients experience severe or moderate physical and psychosocial consequences that can negatively influence both the clinical condition and overall quality of life. The examples of the physical consequences are limitation of the upper limb movements, functional impairment, paresthesia, and lymphedema [2,3,4,5,6,7].

Recent studies show the correlation between the treatment of breast cancer and psychosocial impairments [6,7,8,9]. In addition, those studies demonstrate that the measurement of quality of life related to health becomes important to understand how the psychosocial factors interfere, in general, in the daily activities of the women who underwent breast cancer surgery [5,6,7].

Quality of life (QoL) assessment consists basically of questionnaires. The number of instruments available to assess QoL in cancer patients has increased, and today there are several breast cancer-specific questionnaires in the literature or questionnaires that can be successfully applied to these patients [7,8,9,10,11,12,13,14]. By combining these questionnaires, it can be assumed that the obtained results give a better overview of all aspects of Qol. The World Health Organization Quality of Life-Bref (WHOQOL-BREF) questionnaires have been used to assess general QoL in many studies [15,16,17,18,19,20,21,22,23,24]. The Functional Assessment of Cancer Therapy-Breast (FACT-B) is a QoL questionnaire specific for women with breast cancer. It is derived from the family of the Functional Assessment of Chronic Illness Therapy (FACIT) Measurement System, which represents a collection of health-related quality of life (HRQoL) questionnaires for various chronic diseases [19,25,26,27].

To date, no published studies have been done using these two questionnaires in Serbia simultaneously to assess QoL in women with breast cancer after surgery.

Therefore, the aim of this study was to determine the relationship between the levels of perceived quality of life in patients operated on for breast cancer in relation to the type of surgery, using these two questionnaires.

The secondary objectives were to examine all their various domains in patients operated on for breast cancer at the Clinical Center Nis, Serbia, as well as to perform their analysis in relation to certain given characteristics of patients.

## 2. Methods

### 2.1. Participants

This retrospective cohort study was carried out at the University of Niš, Faculty of Medicine (Serbia) from September 2019 to March 2020. The study included 425 women who underwent breast cancer surgery in the last 25 years. All women were interviewed when they were coming to the control of the Oncology Council of the Clinical Center in Nis, which is also the largest teaching base of the Medical Faculty of the University of Nis.

Participation in the survey was voluntary and anonymous with the prior announcement about the significance of the research and the approval of the management.

The inclusion criterion was women with a primary diagnosis of breast cancer at any stage of the disease, who were submitted to breast cancer surgery in the last 25 years.

The exclusion criterion was previous history of another malignancy.

Out of a total of 438 women interviewed, 13 of them gave incomplete answers, to the extent that they had to be excluded from the sample. Seven women refused to participate in the research.

The study procedures were carried out in accordance with the Declaration of Helsinki, and approvals of the Ethical Committee of the Faculty of Medicine of the University of Niš. Written informed consent was obtained from all participants after the goals of the study and handling of collected data were explained to ensure privacy and confidentiality was understood.

### 2.2. Procedure and Measures

The survey was performed in clinics of Clinical Centre of Niš by the authors. The average time for answering the whole survey was intended to last a maximum of 20 min, including the time needed for instructions.

All of the participants were assessed by using a special survey, which contained overall data, a questionnaire World Health Organization Quality of Life–Bref (WHOQOL-Bref) [26], and a questionnaire Functional Assessment of Cancer Therapy-Breast (FACT-B version 4) [19].

Overall data referred to age, level of education, marital status, kind of surgery, existence of some other illness (comorbidity), and number of years past since surgery.

#### 2.2.1. World Health Organization Quality of Life-Bref (WHOQOL-BREF)

The WHOQOL-Bref questionnaire is an abbreviated version of the WHOQOL-100 [26]. It produces a profile with four domain scores and two individually scored items about an individual’s overall perception of quality of life and health (Score range: 2–10). It contains 26 questions, including two general questions, and the remaining 24 questions represent all aspects of the original instrument. These 24 items are divided into four domains: physical health (questions 3, 4, 10, and 15 to 18)—ranging from 7 to 35, psychological (questions 5, 6, 7, 11, 19, and 26)—ranging from 6 to 30, social relationships (questions 20 to 22)—ranging from 3 to 15, and environment (questions 8, 9, 12 to 14, and 23 to 25)—ranging from 8 to 40.

Each item is rated on a five-point Likert scale (from 1 to 5).

The four domain scores are scaled in a positive direction, with higher scores indicating a higher quality of life. That is why the values of three items (questions 3, 4, and 26) must be reversed before final scoring [25,26].

#### 2.2.2. Functional Assessment of Cancer Therapy-Breast (FACT-B Version 4)

The FACT-B is a breast cancer-specific HRQoL instrument of the FACIT system [27]. The 37-items FACT-B (version 4) are divided into five subscales, namely, physical well-being (PWB) ranging from 0 to 28 (questions GP1 to GP7), social/family well-being (SWB) ranging from 0 to 28 (questions GS1 to GS7), emotional well-being (EWB) ranging from 0 to 24 (questions GE1 to GE6), functional well-being (FWB) ranging from 0 to 28 (questions GF1 to GF7), and the additional concerns for breast cancer—breast cancer subscale (BCS) ranging from 0 to 40 (questions B1 to B9) [19,27].

Each item is rated on a five-point Likert scale (from 0 to 4).

The score is calculated separately for each scale by adding up the points for each question. The higher the score is, the better the patient’s QoL [27]. The values for some questions, which are negatively worded (GP1 to GP7, GE1, GE3 to GE6, B1 to B3, B5 to B8, and B10 to B13) are inverted in the calculation of the final score, so that a higher score indicates a better HRQoL.

The FACT-B total score is the sum of scores of all five subscales (score range: 0–148); the FACT-G score is the sum of PWB, SWB, EWB, and FWB (0–108); and the Trial Outcome Index (TOI) is the sum of scores of the PWB, FWB, and BCS (0–96).

### 2.3. Statistical Analysis

The statistical analysis was performed using the SPSS 17.0 program (SPSS Inc., Chicago, IL, USA) in Windows 7 Ultimate. The research results were presented in tables.

The statistical analysis of the data included the application of descriptive tests, as well as multiple linear regression analysis and correlation tests. The descriptive statistics were performed to report the analysis of the data that were presented as mean and standard deviations. The categorical variables were shown as frequency and percentages.

Pearson and Spearman correlations were used to determine the strength of the relationships between the examined variables. For evaluation of the influence of the examined factors on WHOQOL-Bref domains and FACT-B subscales, we used multiple linear regression analysis. Coefficients of linear regression (β) are calculated and displayed along with their 95% confidence intervals. Evaluation of the statistical significance of the value was performed by T-test. Coefficients represent changes in WHOQOL-Bref and FACT-B scales caused by the value increase independent of variables for one unit of measurement. To further compare the difference in satisfaction between three types of operative procedures, we performed Kruskal–Wallis tests. The statistical significance was set at *p* < 0.05.

## 3. Results

### 3.1. General Characteristics of Participants with WHOQOL-BREF and FACT-B Scores

The study included 425 randomly selected women. Socio-demographic and other examined characteristics of the surveyed women are shown in the Table 1.

Their mean age was 57.83 (SD 11.36). Out of 425 surveyed women, 170 of them (40%) had completed some college or university. There were 303 women who were married or living in a community with an extramarital partner (71.3%), while there were single, separated, divorced, or widowed 122 (28.7%). In addition to breast cancer, 340 women surveyed (80%) were diagnosed with another disease. Two hundred and fifty six examined women (60.2%) underwent mastectomy without reconstruction, 115 of them (27.1%) underwent breast-conserving surgery (quadrantectomy with axillary node dissection and radiation therapy), and 54 (12.7%) women underwent mastectomy with breast reconstruction (Table 1).

Table 2. shows the mean values of the scores of the two questionnaires used to measure the quality of life of patients operated on for breast cancer according to all domains and subscales of these questionnaires (Table 2).

### 3.2. Correlations between Examined Characteristics of the Surveyed Women and HLQoL

The associations between some characteristics of the surveyed women and the domains of health related quality of life (HRQoL) measured by WHOQOL-Bref and FACT-B were examined by a Pearson correlation coefficient (i.e., by Spearman’s rho rank correlation coefficient). Preliminary analyses were conducted in order to check the assumptions of normality, linearity, and homogeneity of the variants. The following correlations were calculated (Table 3).

In WHOQOL-Bref-derived scores, among other significant correlations, it was found that the type of surgery was significantly associated with OQoLaGH (ro = 0.19, *p* < 0.001), then with the aspect of psychological health (ro = 0.23, *p* < 0.001), as well as with the aspect of social health (ro = 0.15, *p* < 0.001).

In scores derived from the FACT-B domains, among other significant correlations, it was found that the type of surgery was significantly associated only with social/family well-being (ro = 0.18, *p* < 0.001) and emotional well-being (ro = 0.10, *p* < 0.05).

### 3.3. Prediction of the Levels of Quality of Life in Order with Type of Breast Cancer Surgery Applied

Regression models were made for the examined parameters that showed a strong correlation with the type of applied operation (with statistical significance of *p* < 0.001). For each of these 4 parameters, these 4 models included, in addition to the type of operation, 5 other characteristics of the examined women (age, education, marital status, existence of another disease (comorbidity), time elapsed since the operation).

First, marital status and type of surgery are recoded into dichotomous variables, where marital status is coded as 1 = not married (single, separated, divorced or widow), 2 = married (married or living together with partner), and type of surgery is coded as 1 = mastectomy, 2 = breast-conserving surgery or mastectomy with breast reconstruction.

Multiple linear regression analysis found that education, existence of comorbidities, time elapsed since surgery, and type of surgery were significant predictors of OQoLaGH (Table 4). Namely, participants’ OQoLaGH (overal quality of life and general health) increased by 0.16 times for each subsequent year of surgery, or by 0.34 times for each subsequent higher education level. Breast-conserving surgery or mastectomy with breast reconstruction was statistically significantly more pronounced (β = 0.18) compared to classical mastectomy.

However, in the other 3 models, multiple linear regression did not single out the type of operation as a statistically significant predictor compared to the other examined characteristics in any of these 3 models (Table 4). In other words, the influence of the type of operation was not decisive in the models with the other five monitored parameters. Other characteristics of the participants took precedence in these cases.

To further compare the difference in women’s satisfaction, standardized according to WHOQOL-Bref and FACT-B questionnaires between the three types of operative procedures, we performed Kruskal–Wallis tests.

Kruskal–Wallis tests revealed statistically significant differences between the three types of surgery in relation to patient satisfaction (Table 5). Breast reconstruction had the worst impact on overall quality of life and general health. For physical health, social relationships, and environment, breast-conserving surgery has proven as best. For social/family well-being, functional well-being, as well as for the total FACT-B score, the breast-conserving surgery also had the best scores (Table 5).

## 4. Discussion

A woman’s breast is not just an organ producing breast feeding milk, but is also a symbol of femininity. Breast loss is the loss of a part of the body and affects the psychosocial experience of losing femininity. Therefore, surgical treatment of breast cancer must be meaningful and adequate in accordance with current oncological treatment protocols and with as few sequelae as possible, in order for patients to continue the best possible quality of life. The surgical approach to in breast cancer surgery is based on TNM classification and stage, the patient’s age, breast volume, and tumor histology. Patient have to be informed and give written consent for surgery. The oncology board must take into account the quality is life after each type of surgery and respect patient’s decision. Patients, especially younger and more educated ones, to whom is made available a lot of internet media, insist on the quality of life and ask for more detailed explanations about particular situation.

Our research confirms that there is a significant difference in the quality of life for patients who underwent breast preservation or breast reconstruction, compared with patients in whom radical mastectomy was performed. The quality of life is especially better in younger patients.

In the study of Manoj et al. back in 2006, it was found that QoL was significantly worse among women after mastectomy compared with breast-conserving surgery [28]. Research by other authors later showed similar results [29,30]. In addition, similar findings are found in other papers, namely, women in all ages after radical mastectomy had the worst level of general satisfaction with life [31,32,33,34,35,36,37,38].

Thanks to improvements in screening, early detection, and surgical and medical treatment, more women are becoming cancer survivors [39,40]. Neoadjuvant and/or postoperative oncological treatments of women with breast cancer clearly improve overall survival. However, those treatments bring the risk of severe side effects and could compromise patients’ wellbeing [41,42]. Consequently, many breast cancer survivors suffer from plenty of long-term, cancer-treatment-related late complications and side effects that impair health-related quality of life (HRQoL), many years after diagnosis and treatment [43,44,45,46,47].

The instruments we used can also be applied in health economic analyses and favor the calculation of quality-adjusted life years (QALY), including scores of reduced morbidity and mortality [48,49]. QALY presents a tool defined as common measure in which diverse outcomes can all be expressed.

Protocols for surgery and adjuvant therapies of breast cancer are well established, but quality of life instruments for these patients after mastectomy and reconstruction are neglected, although the good-quality QoL instruments specific to breast cancer outcomes are useful. These instruments improve our understanding of life quality of women with breast cancer, especially for upper limb disability [47,48].

Living with the sequel of breast cancer treatment has influence on both physical and psychological health. The term survivorship is related to the physical, psychosocial, spiritual, social, and economic consequences of breast cancer left over from acute treatment [50,51,52,53,54,55].

In addition, our experience is that breast-conserving surgery, when performed in women with more massive breast volume, where the remaining breast tissue is sufficient to meet relative symmetry, did not want symmetry with reconstructive surgery. In general, patients have a clearly defined consent for the operation, which states everything about possible complications and the length of treatment. However, when the patient is highly motivated for breast reconstruction, she does not attach importance to all these possible problems.

In women whose breast volume was small, breast-conserving surgery was rarely performed, because the patients were more motivated for mastectomy with or without reconstruction. That is because in our country, the fear of breast cancer is significant, and women often define that removing their breasts, if they are naturally very small, will not be an aesthetic problem—they will feel safer better than aesthetic. It is worth mentioning that the largest number of reconstructions was done as a single stage procedure and there were far fewer of them.

Clinically significant psychological stress after knowing cancer diagnosis is common, and breast cancer patients are even more vulnerable experiencing depression and/or anxiety. The intense psychological stress significantly impairs personal functioning and overall quality of life. The recognition of psychological distress by therapist (surgeon, oncologist) is now defined as a vital component of good clinical care.

According to our experience, open communication with medical staff, partner empathy, and support were all found to increase women’s psychological wellbeing and perception of body image and sexuality. In addition, the interaction between the patient and doctor during clinical consultation has a significant influence on the patient’s overall breast cancer experience, and in the case of good interpersonal communication the overall psychological stress is diminished. Doctors need to be required for master excellent communication skills in order to provide good clinical care, but the patient as a factor also influences the breast cancer patient–doctor relationship.

The most prominent limitation of the study was the relatively small sample. First impact had single university center survey. Second, researchers used a self-report method for data collection. Therefore, it is possible that a certain number of respondents had the desire to “reduce their issues” and therefore overestimate their experiences, which could have caused bias. From this perspective, it could be advisable to use not only the WHOQOL and the FACT-B for QoL measures for breast cancer. Future studies should also consider collecting data from multiple treatment centers and different ways of communicating (e.g., electronic devices).

## 5. Conclusions

Our research confirms that there is an evident difference in the quality of life in patients in whom the breast has been preserved or reconstructed in relation to patients in whom mastectomy has been performed. For most components and domains of the quality of life perceived by patients, breast-conserving has proven as the best type of surgery.

## Figures and Tables

**Table 1 medicina-56-00402-t001:** General characteristics of the participants (N = 425).

Examined Characteristics	Min	Max	Mean	SD
Age	37	80	57.83	11.36
Years after surgery	1	25	7.25	5.31
**Examined Characteristics**			**N (425)**	**% (100)**
Education level	None		18	4.2
	Primary school		69	16.2
	Secondary school		168	39.5
	Some college		45	10.6
	University		125	29.4
Marital status	Single		19	4.5
	Married		301	70.8
	Separated		3	0.7
	Divorced		26	6.1
	Living together		2	0.5
	Widow		74	17.4
Existence of some comorbidity	No		85	20.0
	Yes		340	80.0
Type of breast cancer surgery	Mastectomy		256	60.2
	Breast-conserving		115	27.1
	Mast. with breast reconstruction		54	12.7

SD: Standard Deviation; N: number.

**Table 2 medicina-56-00402-t002:** Overview of the WHOQOL-BREF and FACT-B scores of the participants.

	Min	Max	Mean	SD
**WHOQOL-BREF**				
Overall quality of life and general health	3	10	6.96	1.50
Physical health	8	33	25.81	5.04
Psychological domain	6	30	21.64	4.46
Social relationships	3	15	11.63	2.38
Environment	16	39	29.63	5.57
**FACT-B**				
PHYSICAL WELL-BEING	6	28	21.07	5.41
SOCIAL/FAMILY WELL-BEING	7	28	23.34	4.07
EMOTIONAL WELL-BEING	1	24	17.58	5.66
FUNCTIONAL WELL-BEING	4	28	19.28	5.38
BREAST CANCER SUBSCALE	12	36	26.41	6.17
FACT-B Trial Outcome Index	22	89	66.77	15.03
FACT-G total score	18	107	81.28	17.78
FACT-B total score	30	139	107.69	22.72

FACT-B: Functional Assessment of Cancer Therapy-Breast; WHOQOL-BREF: The World Health Organization Quality of Life-Bref.

**Table 3 medicina-56-00402-t003:** Correlations between examined characteristics and WHOQOL-BREF and FACT-B scores.

	Age	Education Level	Marital Status	Existence of Comorb.	Years after Surgery	Type of Surgery
OQoLaGH	−0.010	0.313 **	−00.022	−00.355 **	0.237 **	0.190 **
*p*	0.834	0.000	0.656	0.000	0.000	0.000
Physical Health	−0.341 **	0.347 **	0.202 **	−0.309 **	0.091	0.070
*p*	0.000	0.000	0.000	0.000	0.060	0.150
Psychological	−0.260 **	0.460 **	0.060	−0.251 **	0.131 **	0.234 **
*p*	0.000	0.000	0.220	0.000	0.007	0.000
Social relationships	−0.330 **	0.309 **	0.535 **	0.013	0.145 **	0.148 **
*p*	0.000	0.000	0.000	0.789	0.003	0.002
Environment	−0.088	0.365 **	0.187 **	−0.147 **	0.207 **	0.093
*p*	0.070	0.000	0.000	0.002	0.000	0.056
PWB	−0.144 **	0.364 **	−0.014	−0.410 **	0.075	−0.029
*p*	0.003	0.000	0.776	0.000	0.125	0.548
SWB	−0.368 **	0.216 **	0.434 **	−0.283 **	0.153 **	0.179 **
*p*	0.000	0.000	0.000	0.000	0.002	0.000
EWB	−0.338 **	0.333 **	−0.091	−0.426 **	0.055	0.097 *
*p*	0.000	0.000	0.060	0.000	0.257	0.046
FWB	−0.190 **	0.428 **	0.120 *	−0.251 **	0.217 **	0.061
*p*	0.000	0.000	0.014	0.000	0.000	0.206
BCS	−0.253 **	0.383 **	−0.081	−0.378 **	−0.110 *	0.073
*p*	0.000	0.000	0.096	0.000	0.024	0.131
FACT-B total score	−0.298 **	0.414 **	0.053	−0.432 **	0.080	0.014
*p*	0.000	0.000	0.273	0.000	0.098	0.768

Notes: ** Correlation is significant at the 0.01 level, and * at the 0.05 level. OQoLaGH—overal quality of life and general health; Physical Health—physical component of health; Psychological—psychological component of health; Social relationships—social component of health; Environment—environmental component of health (measured by WHOQOL-Bref). PWB—physical well-being; SWB—social/family well-being; EWB—emotional well-being; FWB—functional well-being; BCS—breast cancer subscale; Trial Outcome Index—TOI (measured by FACT-B); Comorb.—Comorbidities.

**Table 4 medicina-56-00402-t004:** Results of the multiple linear regression analysis—assessing the relationships between World Health Organization Quality of Life-Bref (WHOQOL-Bref) and Functional Assessment of Cancer Therapy-Breast (FACT-B) scores with the observed parameters.

Characteristics	Unstandardized Coefficients		Standardized Coefficients	t	*p*	95% CI		Collinearity Statistics	
Model	B	Std. Error	β			Lower	Upper	Tolerance	VIF
OVERALL QUALITY OF LIFE AND GENERAL HEALTH				R^2^ = 0.251; F = 23.371, df = 6, *p* < 0.001					
Constant	6.12	0.63		9.720	0.000	4.88	7.35		
Age	0.002	0.01	0.013	0.261	0.794	−0.01	0.02	0.71	1.41
Education	0.43	0.060	0.340	7.188	0.000	0.31	0.55	0.80	1.25
Marital status	0.31	0.159	0.093	1.943	0.053	−0.01	0.62	0.78	1.28
Comorbidity	−1.02	0.163	−0.271	−6.232	0.000	−1.34	−0.70	0.95	1.06
Years since surgery	0.045	0.013	0.158	3.336	0.001	0.02	0.07	0.80	1.26
Type of surgery	0.554	0.150	0.181	3.702	0.000	0.26	0.85	0.75	1.33
DEPENDENT VARIABLE: PSYCHOLOGICAL DOMAIN				R^2^ = 0.548; F = 29.921, df = 6, *p* < 0.001					
Constant	18.79	1.805		10.406	0.000	15.237	22.334		
Age	−0.058	0.019	−0.147	−3.027	0.003	−0.095	−0.020	0.708	1.413
Education	1.422	0.171	0.379	8.300	0.000	1.085	1.758	0.801	1.248
Marital status	0.893	0.455	0.091	1.962	0.050	−0.002	1.788	0.784	1.275
Comorbidity	−2.322	0.469	−0.208	−4.954	0.000	−3.244	−1.401	0.946	1.057
Years since surgery	0.122	0.039	0.145	3.165	0.002	0.046	0.198	0.796	1.256
Type of surgery	0.527	0.430	0.058	1.227	0.221	−0.318	1.372	0.752	1.330
SOCIAL RELATIONSHIPS DOMAIN				R^2^ = 0.441; F = 55.027, df = 6, *p* < 0.001					
Constant	6.019	0.860		6.996	0.000	4.328	7.710		
Age	−0.020	0.009	−0.097	−2.229	0.026	−0.038	−0.002	0.708	1.413
Education	0.559	0.082	0.280	6.846	0.000	0.398	0.719	0.801	1.248
Marital status	2.840	0.217	0.541	13.092	0.000	2.414	3.266	0.784	1.275
Comorbidity	−0.468	0.223	−0.079	−2.096	0.037	−0.907	−0.029	0.946	1.057
Years since surgery	0.035	0.018	0.078	1.903	0.058	−0.001	0.071	0.796	1.256
Type of surgery	0.084	0.205	0.017	0.412	0.681	−0.318	0.487	0.752	1.330
SOCIAL/FAMILY WELL-BEING SUBSCALE				R^2^ = 0.391; F = 44.700, df = 6, *p* < 0.001					
Constant	22.36	1.538		14.535	0.000	19.332	25.378		
Age	−0.086	0.016	−0.239	−5.257	0.000	−0.118	−0.054	0.708	1.413
Education	0.482	0.146	0.141	3.300	0.001	0.195	0.769	0.801	1.248
Marital status	3.609	0.388	0.401	9.307	0.000	2.847	4.372	0.784	1.275
Comorbidity	−2.970	0.399	−0.292	−7.436	0.000	−3.755	−2.185	0.946	1.057
Years since surgery	0.114	0.033	0.148	3.463	0.001	0.049	0.178	0.796	1.256
Type of surgery	−0.253	0.366	−0.030	−0.692	0.490	−0.973	0.466	0.752	1.330

Notes: β—Beta coefficient in regression ANOVA analysis of potential predictors; CI—confidence interval.

**Table 5 medicina-56-00402-t005:** Kruskal–Wallis tests—type of operation in relation to satisfaction parameters according to WHOQOL-Bref and FACT-B questionnaires.

	Radical Mastectomy (N = 256)	Breast-Conserving (N = 115)	Mastectomy with Breast Reconstruction (N = 54)	Chi-Square	*p*
**Mean Rank**					
**WHOQOL-Bref**					
Overall Quality of Life and General Health	231.13	190.47	175.04	15.915	<0.01
Physical health	206.05	239.50	189.53	8.203	0.017
Psychological health	189.90	230.91	284.35	30.182	<0.01
Social relationships	198.49	269.10	162.34	37.949	<0.01
Environment	203.79	252.84	171.79	19.777	<0.01
**FACT-B**					
PHYSICAL WELL-BEING	215.90	216.51	191.78	1.869	0.393
SOCIAL/FAMILY WELL-BEING	195.31	281.82	150.31	56.467	<0.01
EMOTIONAL WELL-BEING	222.61	197.81	199.80	3.996	0.136
FUNCTIONAL WELL-BEING	206.90	240.76	182.81	9.843	0.007
BREAST CANCER SUBSCALE	205.71	258.62	150.44	30.967	<0.01
FACT-B total score	211.57	242.17	157.66	17.511	<0.01

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
