# Peer review of "Quality of Life Indicators in Patients Operated on for Breast Cancer in Relation to the Type of Surgery—A Retrospective Cohort Study of Women in Serbia"

_medicina, 2020, doi:10.3390/medicina56080402_

Round 1
Reviewer 1 Report
83/5000 I would suggest to correct the sentence line 298 "A woman's breast is not just an organ for making breast milk, but it is an important secondary sexual characteristic." using words more suitable for scientific paper I am satisfied with the review carried out
Author Response
Dear reviewer,
Thank you very much for your kind suggestion. We have corrected the underlined sentence and it is now:
A woman's breast is not just an organ producing breast feeding milk, but is also a symbol of femininity.
Reviewer 2 Report
1. In addition, our experience is that sparing breast surgery when performed in women with more massive breast volume, where the remaining breast tissue is sufficient to meet relative symmetry, did not want symmetry with reconstructive surgery. In women whose breast volume was small, sparing surgery was rarely performed, because the patients were more motivated for mastectomy and reconstruction or refused breast reconstruction.
This paragraph needs to be impproved.
Have the patients been told that the operated breast will be irradiated and that the volume of the latter will be even smaller after radiation therapy? An explanation must be given when choosing not to mirror the opposite breast. What does sparing surgery mean? Is it a skin sparing mastectomy or a breast conserving surgery?
Why do patients prefer a mastectomy over breast conserving surgery for small breasts? The majority of patients do not prefer mastectomy if it is explained to them that the prognosis is the same for breast conserving surgery with radiotherapy. This must be explained. 2. In the conclusion, it is necessary to detail the difference in quality of life between a Breast-conserving surgery or mastectomy with breast reconstruction. Indeed, a breast reconstruction often involves more than a single intervention, more time off work, more pain. Does all of this really have no influence on patient satisfaction and quality of life? Doesn't breast-conserving surgery have a better impact on the quality of life of patients?
Author Response
Dear reviewer,
We did our best to further improve the mentioned paragraph with a few more sentences added answering your valuable questions.
“1. In addition, our experience is that sparing breast surgery when performed in women with more massive breast volume, where the remaining breast tissue is sufficient to meet relative symmetry, did not want symmetry with reconstructive surgery. In women whose breast volume was small, sparing surgery was rarely performed, because the patients were more motivated for mastectomy and reconstruction or refused breast reconstruction.
This paragraph needs to be impproved.
Have the patients been told that the operated breast will be irradiated and that the volume of the latter will be even smaller after radiation therapy? An explanation must be given when choosing not to mirror the opposite breast. What does sparing surgery mean? Is it a skin sparing mastectomy or a breast conserving surgery?
Why do patients prefer a mastectomy over breast conserving surgery for small breasts? The majority of patients do not prefer mastectomy if it is explained to them that the prognosis is the same for breast conserving surgery with radiotherapy. This must be explained.”
This whole paragraph now looks as this:
In addition, our experience is that breast-conserving surgery when performed in women with more massive breast volume, where the remaining breast tissue is sufficient to meet relative symmetry, did not want symmetry with reconstructive surgery. In general, patients have a clearly defined consent for the operation, which states everything about possible complications and the length of treatment. However, when the patient is highly motivated for breast reconstruction, she does not attach importance to all these possible problems.
In women whose breast volume was small, breast-conserving surgery was rarely performed, because the patients were more motivated for mastectomy with or without reconstruction. That is because in our country, the fear of breast cancer is significant, and women often define that removing their breasts, if they are naturally very small, will not be an aesthetic problem - they will feel safer better than aesthetic. By the way, the largest number of reconstructions was done as a single stage procedure and there were far fewer of them.
“2. In the conclusion, it is necessary to detail the difference in quality of life between a Breast-conserving surgery or mastectomy with breast reconstruction. Indeed, a breast reconstruction often involves more than a single intervention, more time off work, more pain. Does all of this really have no influence on patient satisfaction and quality of life? Doesn't breast-conserving surgery have a better impact on the quality of life of patients?”
In the Conclusions we also added a new sentence, and now it is:
Our research confirms that there is an evident difference in the quality of life in patients in whom the breast has been preserved or reconstructed in relation to patients in whom mastectomy has been performed. For the most components and domains of the quality of life perceived by patients, breast-conserving has proven as the best type of surgery.
We really hope that we well understood your comments, and that you will find our upgrades of the Discussion and Conclusions sections in accordance with your stated questions.
Kind regards
This manuscript is a resubmission of an earlier submission. The following is a list of the peer review reports and author responses from that submission.
Round 1
Reviewer 1 Report
- the study describes the retrospective experience of one Serbian breast center: please modify the title: a retrospective cohort study of women in a Serbian breast centre
- line 60 (I'd avoid to refer to axillary dissection, that is indipendently linked to the treatment of the breast)
- in references I'd add these three recent and relevant articles:
Bjelic-Radisic V, Cardoso F, Cameron D, et al. An international update of the EORTC questionnaire for assessing quality of life in breast cancer patients: EORTC QLQ-BR45 [published correction appears in Ann Oncol. 2020 Apr;31(4):552]. Ann Oncol. 2020;31(2):283-288. doi:10.1016/j.annonc.2019.10.027
Ghilli M, Mariniello MD, Camilleri V, et al. PROMs in post-mastectomy care: Patient self-reports (BREAST-Q™) as a powerful instrument to personalize medical services. Eur J Surg Oncol. 2020;46(6):1034-1040. doi:10.1016/j.ejso.2019.11.504
Zhang J, Yao YF, Zha XM, Pan LQ, Bian WH, Tang JH. Development and Evaluation of a Patient-Reported Outcome (PRO) Scale for Breast Cancer. Asian Pac J Cancer Prev. 2015;16(18):8573-8578. doi:10.7314/apjcp.2015.16.18.8573
- Years after surgery: the range is too large (1-25) and the mean results quite distant from the surgery if you want to evaluate the QoL after surgery. Moreover it makes the sample not homogeneous. surgery today is too different from 20 years ago. why don't eliminate the oldest cases?
- line 268: please modify in "the OQoLaGH in breast conserving surgery..."
- modify line 268 in a way suitable for a scientific article
Reviewer 2 Report
A description is lacking concerning the overall context of breast reconstruction in Serbia, whether or not insurance is provided for breast reconstruction. Is a reconstruction offered to patients every time, can it be done immediately or not? The different types of mastectomies must be detailed: Skin sparing, nipple sparing or not, immediate or deferred. Overall, the general characteristics of the patients are well described. On the other hand, it is necessary to differentiate from the point of view of satisfaction: Breast conservating surgery and mastectomy with breast reconstruction because these are two completely different types of reconstruction. These two types of surgery should be separated into two different groups. Indeed the group: breast conservative surgery probably has a different satisfaction rate from the group: mastectomy with reconstruction. In addition, it is not clear whether patients who had breast conserving surgery had an oncoplasty. The type of complete reconstruction must also be described: prosthetic vs autologous
(DIEP, latissimus dorsi, SGAP). A paragraph regarding complications and satisfaction rate should be developed